# Cell Salvage at the ICU

**DOI:** 10.3390/jcm11133848

**Published:** 2022-07-02

**Authors:** Stephan L. Schmidbauer, Timo F. Seyfried

**Affiliations:** 1Department of Anesthesiology, University Hospital Regensburg, 93053 Regensburg, Germany; stephan.schmidbauer@ukr.de; 2Department of Anesthesiology, Ernst von Bergmann Hospital, 14467 Potsdam, Germany

**Keywords:** Patient Blood Management, anemia, cell salvage, ICU

## Abstract

Patient Blood Management (PBM) is a patient-centered, systemic and evidence-based approach. Its target is to manage and to preserve the patient’s own blood. The aim of PBM is to improve patient safety. As indicated by several meta-analyses in a systematic literature search, the cell salvage technique is an efficient method to reduce the demand for allogeneic banked blood. Therefore, cell salvage is an important tool in PBM. Cell salvage is widely used in orthopedic-, trauma-, cardiac-, vascular and transplant surgery. Especially in cases of severe bleeding cell salvage adds significant value for blood supply. In cardiac and orthopedic surgery, the postoperative use for selected patients at the intensive care unit is feasible and can be implemented well in practice. Since the retransfusion of unwashed shed blood should be avoided due to multiple side effects and low quality, cell salvage can be used to reduce postoperative anemia with autologous blood of high quality. Implementing quality management, compliance with hygienic standards as well as training and education of staff, it is a cost-efficient method to reduce allogeneic blood transfusion. The following article will discuss the possibilities, legal aspects, implementation and costs of using cell salvage devices in an intensive care unit.

## 1. Introduction

Postoperative anemia is a common problem in surgical intensive care units [1]. Patient Blood Management (PBM) is a possible therapeutic approach to engage this issue. In a new global definition, PBM is described as a patient-centered, systemic and evidence-based approach. Its target is to manage and to preserve the patient’s own blood. This leads to an improvement in patient safety [2]. The transfusion of allogeneic blood products should be considered as a short-time therapy, which is not capable of addressing the root of the problem [2]. The main goal of PBM is to improve the patient’s prognosis and outcome by reducing allogenic transfusion [2]. The use of autologous blood provided by cell salvage procedures has an important role in the intra- and postoperative setting. Washed and filtered autologous blood can be part of the second pillar of PBM and the transition to the third pillar.

Several studies have shown that the use and administration of autologous (washed) blood collected and processed by cell salvage devices can reduce hospital stay and mortality [2,3,4,5]. Using of cell salvage can significantly reduce usage of allogeneic blood intra- and postoperatively in cardiac surgery [6,7,8]. Moreover, the use of autologous blood has reduced postoperative wound infections [3]. In addition, there is a significant cost reduction due to the elimination of testing, processing and storage of donor blood [9]. Furthermore, a 2016 study demonstrated that the function of erythrocytes in autologous, washed blood is better than in stored foreign blood. Moreover, it has been demonstrated that the administration of allogeneic blood increases the risk of postoperative infection [10,11]. acute lung injury [12] as well as a perioperative myocardial infarction [13]. Additionally, allogeneic blood transfusion causes an increase of the 5-year mortality rate in cardiac bypass surgery [14]. 

The use of cell salvage devices in tumor operations is also common and considered safe. In Germany, the reprocessed blood must be irradiated with a dose of 50 Gy before retransfusion [15]. In other countries, such as Great Britain, retransfusion is carried out via a leukocyte depletion filter [16]. This also complies with the guidelines of the European Society of Anesthesiology, for example [17,18].

For the postoperative period, there is the possibility of draining wound blood via drains, collecting its content and retransfusing it to the patient. The collected blood can be returned to the patient unwashed or after washing with a cell salvage device. Transfusion from wound drainage and blood collected intraoperatively or postoperatively without prior washing is not recommended because of the risk of coagulation activation, cytokine and possibly endotoxin leaching and the transfer of other biologically active substances [15,19,20,21,22].

Transfusion should not be the first line of treatment for anemia or blood loss, according to the literature. Instead, mounting data suggest that a proactive, patient-centered approach to managing a patient’s own blood should be the new standard of care.

As a result, an unfavorable transfusion occurrence that could have been avoided with proper Patient Blood Management could be considered a case of medical professional medical negligence. Transfusion medicine practice culture has to transition toward a Patient Blood Management strategy in order to maximize patient safety, with hospitals implementing it as an important tool to reduce the dangers of allogeneic blood transfusion.

In the following, the possibility of cell salvage in the postoperative setting during the intensive care treatment of patients will be discussed along with possible problems and costs associated with this procedure.

## 2. Legal Aspects and Problems of Collection Time

Due to different national regulations, no general recommendation can be made regarding collection time and retransfusion. The German guidelines do not approve the indication-related retransfusion of intraoperatively collected and processed wound blood after a storage period of more than 6 h after initiation of the collection process. Furthermore, it is also recommended that these red cell concentrates prepared with cell savage devices be stored at +2–+6 °C and separate from homologous products until retransfusion. RBC concentrates prepared this way should be retransfused as soon as possible. In special cases, they may be stored for up to 6 h [15]. These recommendations are similar to those of the Austrian and Swiss professional societies [23,24]. The British guidelines even point out that the storage temperature of blood must be in the range of 2 to 6 degrees Celsius, and if the requirement is met, the blood must be transfused within 4 h [25]. The guidelines of the Austrian and Swiss professional societies contain the same recommendation [23,24]. The processing of the collected blood should be started as soon as enough has been collected. Immediately thereafter, retransfusion of the autologous blood should be started. It is recommended that the retransfusion itself should not take longer than 4 h mainly due to the possible growth of microorganisms in uncooled blood [16,26].

The American guidelines and recommendations do not go into more detail. However, they clearly state transfusion triggers for the transfusion of allogeneic blood. For example, hemodynamically stable adult patients should only be transfused from a hemoglobin value of 7 g/dL. Patients with orthopedic and cardiac surgery as well as patients with previous cardiac disease should be transfused from a hemoglobin value of 8 g/dL [27]. This is also in line with most European recommendations, such as the German, British and Austrian recommendations [15,23,25].

These different national recommendations make it difficult to make a global commitment to the use of cell salvage devices and the retransfusion of autologous blood. Nevertheless, it has been demonstrated that these PBM techniques help to minimize allogeneic blood transfusions and the associated costs and risks [28].

In a study conducted by the University of Rochester in 2013, wound blood was collected postoperatively via drains at the patient’s bedside after cardiac surgery in children. It was processed and retransfused when predefined transfusion triggers occurred [29]. The blood was collected over a period of 24 h and cooled to 1–6 °C according to the specifications of the New York blood bank. There were no postoperative complications such as infections, wound healing disorders or immunological reactions. In addition, the mortality rate was significantly lower than in patients who had received more allogenic red blood cell concentrates (RBCs) during their stay.

Many other studies have shown that the retransfusion of postoperatively collected and processed wound blood helps to save allogeneic blood and to minimize the associated complications described above [6,19,30,31,32,33].

In this case, the collected wound blood was either continuously processed and retransfused, or after the occurrence of certain transfusion triggers such as a drop in the hemoglobin below a previously defined limit, increasing catecholamine requirement, in the case of volume deficiency or decrease in cardiac output. 

Especially in cardiac surgery, this approach offers the possibility of a quick and adequate volume supply in the event of complications such as severe postoperative bleeding with rethoracotomies. This can help to stabilize the patient until further allogenic blood products arrive from the blood depot. It was also shown that the rate of reinterventions and the number of foreign blood transfusions were significantly lower when collected wound blood was retransfused [34,35].

## 3. Logistics and Practical Procedure

The following procedure has been established at the University Hospital Regensburg for the postoperative care of patients in cardiac surgery. Operations include coronary bypass surgery, heart valve surgery, aortic surgery, heart transplants and major lung surgery. Wound blood is collected intraoperative using cell salvage devices (Cell Saver Elite, Haemonetics, Boston, MA, USA), processed and, if necessary, retransfused during or at the end of the surgical procedure. At the end of the surgery, the wound drains inserted by the surgeons are connected to a new cell salvage reservoir, which is transferred to the intensive care unit together with the patient. Similar to conventional chest drains, it is connected to a vacuum suction system (−20 mmgHg). This is shown in Figure 1. The wound blood collected here is measured and, in the event of more severe bleeding or increasing hemodynamic instability of the patient, is washed with an autotransfusion device and returned to the patient. As recommended by the device producers, the collected blood is mixed with a heparin saline mixture to make it non-coagulable. For this purpose, 25,000 IU of heparin are added to one liter of 0.9% sodium chloride solution. This solution is washed out when the collected blood is processed. In certain cases, it can even happen after more than 6 h of collection time after an individual risk–benefit assessment. Figure 2 helps to get an overview. The internal monthly quality controls of the used autotransfusion device on the cardiac intensive care unit show a sufficient removal of protein (<90%) and a hematocrit above 50%. Potassium was also reduced by more than 90%. 

An autotransfusion device is prepared and ready for use at all times for emergency situations at the intensive care unit. In order to save resources in terms of autotransfusion equipment, the devices are exchanged every two days and the device that has already been set up for use is brought to the operating theatre and then replaced by another device with a fresh set. 

This is particularly important for the mandatory quality controls of the devices used. Blood samples are taken at regular intervals from the reservoir before washing and from the product after the washings process. Samples are tested for total protein, albumin and hematocrit in order to check the correct function of the equipment (the threshold for the plasma elimination rate is >90%). The plasma elimination rates are calculated according to the following equation: Elimination rate (%) = 100 − 100 × ((V_WRBC_ × (1 − Hct_WRBC_/100) × C_WRBC_)/(V_TB_ × (1 − Hct_TB_/100) × C_TB_)), where V_TB_ × (1 − Hct_TB_/100) is the volume of the supernatant in the test blood, while V_WRBC_ × (1 − Hct_WRBC_/100) represents the volume of the supernatant in the produced washed RBC (mL). C_WRBC_ and C_TB_ are the concentrations of the respective substance in the supernatant of the produced washed RBCs [36]. 

Furthermore, the hematocrit has to be checked once during each cell salvage application (threshold > 50%). 

On the morning of the first post-operative day, the reservoirs are exchanged for conventional chest drainage containers and the unneeded collected wound blood is discarded. 

Strict adherence to hygienic standards as well as regular training and education of the staff in the use of autotransfusion devices is essential. Especially in case of emergency, the staff needs to be trained well to handle the provided equipment. 

The Serious Hazards of Transfusion from 2020 recommends the following steps related to education and training [37]:Training and competency evaluations of cell salvage personnel are mandatory (documented in the form of a training record);An individual’s identification number and unique case number must be written on every bag of cell salvage blood;Whenever intraoperative cell salvage (ICS) or postoperative cell salvage (PCS) is performed, SHOT should be notified of any adverse events;As important as monitoring patients for the reinfusion of allogeneic red cells is monitoring patients for red cells collected via ICS or PCS;The SHOT Reports from previous years should be reexamined by practitioners, in particular those that deal with autologous transfusion, to ensure that historic incidents do not repeat themselves. Action: Cell salvage teams.

## 4. Costs

The use of cell salvage is associated with certain costs. Due to the higher quality and safety compared to unwashed wound blood, these costs can be justified. The prices for the sets for collection or the reservoir varies from 26 to 41 EUR, depending on the manufacturer. The costs for a complete set for reprocessing and retransfusion amount to about 80 to 100 EUR according to the manufacturers Haemonetics^®^ Cell Saver^®^ Elite^®^+ Autotransfusion System and the Xtra^®^ Autotransfusion System by LivaNova^®^ (Xtra, LivaNova, London, UK) in Germany.

The costs of transfusing a red blood cell concentrate show a big variety from country to country. In Germany, for example, the price of one pack of allogenic blood is costs about 140 EUR [38], in Austria approximately 150 EUR (Source Austrian Red Cross Status 2020) [39], in the Netherlands 180 EUR [40] and in Italy around 130–215 EUR [41]. The British NHB estimates the average cost of an RBC to be around 165 EUR, plus the cost of in-clinic testing and administration [42]. In the USA, it is priced at a converted rate of about 326 EUR per RBC [43]. In France, the transfusion of a red blood cell concentrate costs 340 EUR [44] and in Canada the equivalent of 574 EUR [45]. 

The abovementioned amounts reflect the cost directly associated with the transfusion of allogenic blood. Since the retransfusion of allogenic blood is associated with increased wound infection and a longer hospital stay, the indirect costs of allogenic have to be taken into consideration [3,4,5,6]. 

The remuneration for the administration of red blood cell concentrates in Germany is regulated in the DRG (Diagnosis Related Groups) flat rate catalogue via the additional charge 107 “Administration of RBCs” and is staggered according to the quantity of RBCs transfused. An additional charge for the administration of RBCs in adult patients is only triggered from a transfusion of six RBC units. If fewer RBCs are transfused, the remuneration is covered by the corresponding flat rate per case and does not lead to any additional revenue [46].

The use of cell salvage—also outside the operating theatre—offers potential for reducing costs and resources in the sense of allogeneic red cell concentrates [38]. Patient Blood Management and, specifically, autotransfusion can also help to reduce the length of hospital stay, and thus, also save further costs for health insurance funds [4]. 

## 5. Discussion

In total, 50–60% of all transfusions take place in the surgical setting [47]. Cell salvage is an effective and cost-effective way to avoid allogeneic blood transfusions intraoperatively and postoperative in intensive care medicine. The application in orthopedic, trauma and cardiac surgery patients seems to be particularly well studied and feasible. These disciplines also have the highest transfusion rates. 

Several large studies after orthopedic surgery have shown that reprocessing and retransfusion of wound blood helped to avoid allogeneic blood transfusions and minimized the risks associated with them. The amount of allogeneic blood used could be reduced by 20–40%. This also had positive effects on hospital length of stay and discharge rates [31,32,48,49,50,51].

In total, 10% of all allogenic RBCs are used in cardiothoracic surgery [35]. For cardiac surgery, it has also been shown that the use of cell salvage devices helps to reduce allogeneic blood transfusions by 30–40% and even to reduce the rate of reinterventions. Moreover, the number of wound infections in the sternal region was not increased and the amount of post-operative bleeding after retransfusion of reprocessed drainage blood was lower [6,8,29,34].

There are currently only limited data and recommendations for the postoperative area and the use of autotransfusion devices in intensive care units or in the recovery room. The evidence, however, is sufficient to extend the use of cell salvage to the ICU for certain situations.

## 6. Summary

In summary, cell salvage is an important component of Patient Blood Management. The use of cell salvage devices is also feasible in the intensive care unit and helps to save allogeneic blood. This might help to minimize the risks associated with the transfusion of allogeneic blood and also to reduce costs. However, compliance with hygienic standards, continuous quality control and regular training of staff are essential for the efficiency of this technique.

## Figures and Tables

**Figure 1 jcm-11-03848-f001:**
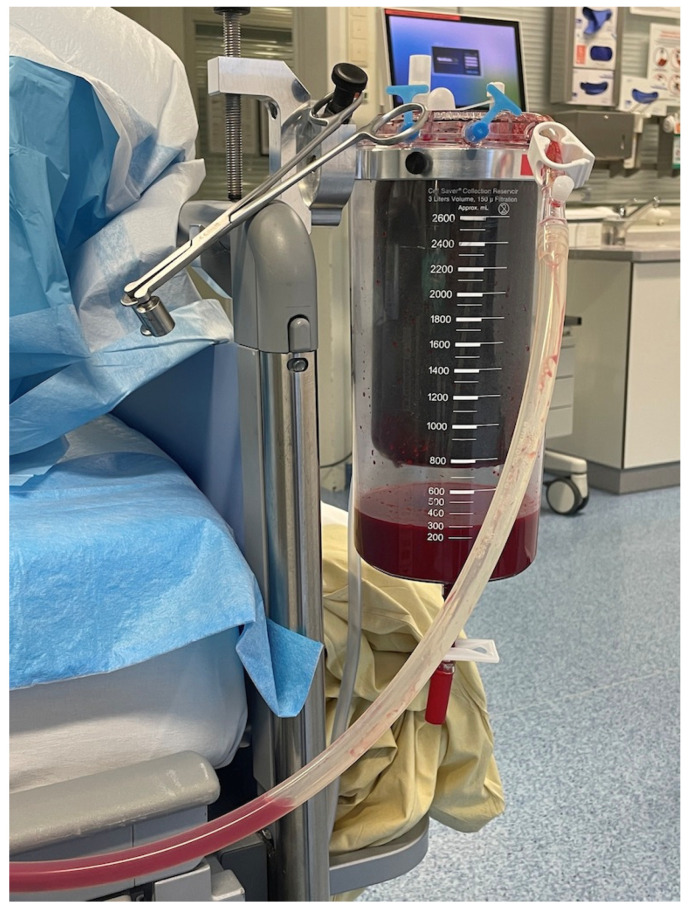
Cell saver reservoir at the bedside of an intensive care patient.

**Figure 2 jcm-11-03848-f002:**
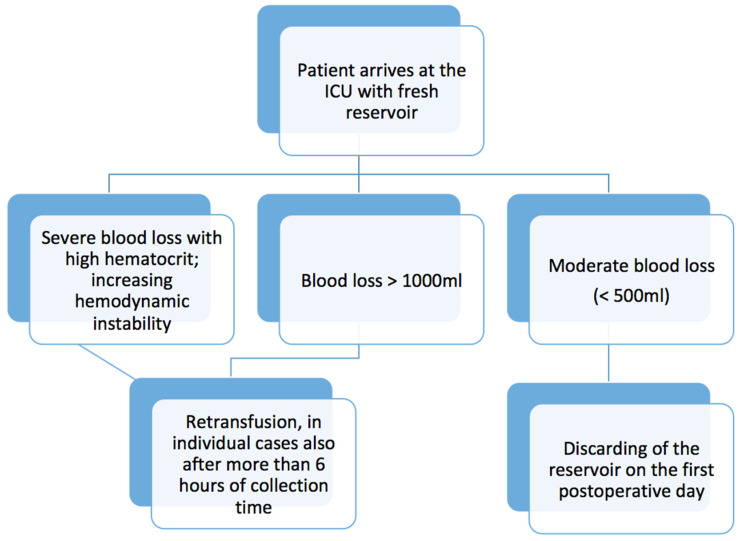
Flowchart for autotranfusion at the ICU.

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
