# Peer review of "Cell Salvage at the ICU"

_jcm, 2022, doi:10.3390/jcm11133848_

Round 1
Reviewer 1 Report
Dear authors,
Thank you very much for your hard work in preparing and conducting this research. Indeed this is an interesting topic. However, there are some proposals to major changes of your manuscript. In Introduction there will be more precise definition of PBM. Cell saving is one of methods, but only introduction of all parts can have full beneficial potential. In chapter 2 you insist on storage of washed RBC from cell saver. It is more logical to return to patient as soon as RBC are processed regardless to Hb trigger, because of blood loss. What about anticoagulation? Describe what drug do you use (UFH or citrate), what dose etc. Chapter RESULTS is missing. It would be better to write of your cases, to give some numbers, eg. how many ml washed and returned postop, in what kind of surgeries do you work in during timeframe. Chapter Costs- it is very difficult to analyze cost of hospital stay and medical procedures. Of course that cell saver and its components have a price as product. But general cost of hospital stay is connected to all medical procedures including complications concerning blood transfusion. So it is not easy to compare price of allogeneic RBC and price of RBC from cell saver.
With grammar corrections and better explanation of your work this can be useful article. In attachment is word file with proposal of corrections.
Best regards.

Author Response
Dear Reviewer
Please find attached the revised version of our article.
Best regards
Stephan Schmidbauer

Reviewer 2 Report
This is a look at feasibility of cell recovery (the current term for cell salvage) in the ICU.
To begin, the first paragraph describing PBM is inaccurate and old. Please revisit with the e-pub 2022, A&A "Global definition of PBM".
The authors could use a table with Pro vs Con for cell recovery in the iCU. Many cardiac surgery center have abandoned this procedure for various reasons. In some situation, the argument is made that treat anemia preoperative and using less phlebotomy (not addressed by the authors) would make cell recovery useless.
In 2003 publication Surgery Without Blood, Shander A. CCM 2003, the table shows (referenced) the number of units "saved" with cell recovery.
Author Response
Dear Reviewer
thank you very much for your feedback!
Please find attached the revised version of our article.
Best regards
Stephan Schmidbauer

Reviewer 3 Report
I have carefully read this interesting article which I think is impotent for readers internationally. In particular, although Patient Blood Management (PBM) is now very well known, the focus is often more on the first pillar but also the second pillar, as shown in this article can become particularly incisive. I believe the article deserves publication, however I can suggest some short improvements:
- the part inherent to the legal aspects is interesting, the authors were very correct in declaring that "Due to different national regulations, no general recommendation can be made regarding collection time and retransfusion", however it could be useful to remember that there are medico-legal aspects important for the implementation of the PBM and that it currently represents a real and useful element for reducing risks in transfusion medicine (doi: 10.1016 / j.transci.2020.102779.)
- the authors explain very well how cell savage is useful in many fields, however many doctors still do not use it in case of suspected oncological pathology and think that there may be a risk of dissemination. Could you include a little reflection on this?
Author Response

(The authors gave the same response as above.)

Round 2
Reviewer 3 Report
I believe that the authors have improved the text according to the indications received. I recommend the publication